# Organometallic compounds as carriers of extraterrestrial cyanide in primitive meteorites

Karen E. Smith[1,2], Christopher H. House [2], Ricardo D. Arevalo Jr.[3], Jason P. Dworkin [4,5] & Michael P. Callahan[1,4,5]

Extraterrestrial delivery of cyanide may have been crucial for the origin of life on Earth since cyanide is involved in the abiotic synthesis of numerous organic compounds found in extant life; however, little is known about the abundance and species of cyanide present in meteorites. Here, we report cyanide abundance in a set of CM chondrites ranging from $50 \pm 1$ to $2472 \pm 38$ $nmol \cdot g^{-1}$, which relates to the degree of aqueous alteration of the meteorite and indicates that parent body processing influenced cyanide abundance. Analysis of the Lewis Cliff 85311 meteorite shows that its releasable cyanide is primarily in the form of $[Fe^{II}(CN)_5(CO)]^{3-}$ and $[Fe^{II}(CN)_4(CO)_2]^{2-}$. Meteoritic delivery of iron cyanocarbonyl complexes to early Earth likely provided an important point source of free cyanide. Iron cyanocarbonyl complexes may have served as precursors to the unusual $Fe^{II}(CN)(CO)$ moieties that form the catalytic centers of hydrogenases, which are thought to be among the earliest enzymes.

[1] Department of Chemistry and Biochemistry, Boise State University, Boise, ID 83725, USA. [2] Department of Geosciences and Penn State Astrobiology Research Center, Pennsylvania State University, University Park, PA 16801, USA. [3] Department of Geology, University of Maryland, College Park, MD 20742, USA. [4] Goddard Center for Astrobiology, NASA Goddard Space Flight Center, Greenbelt, MD 20771, USA. [5] Astrochemistry Laboratory, NASA Goddard Space Flight Center, Greenbelt, MD 20771, USA. Correspondence and requests for materials should be addressed to M.P.C. (email: michaelcallahan914@boisestate.edu)

Carbonaceous chondrites are fragments of ancient asteroids that provide a record of the chemistry of the early solar system[1,2]. They contain a variety of organic compounds and their delivery to early Earth may have played an essential role in the chemistry that led to the origin of life[3–11]. Hydrogen cyanide (HCN) has been detected in water extracts of the Murchison meteorite upon acidification[12], which was a surprising result because cyanide is a highly reactive compound and thought to have been completely consumed by reactions in the parent asteroid. Isotope ratio mass spectrometry indicated that this cyanide was extraterrestrial in origin; however, low $^{13}$C enrichment and low $^{15}$N enrichment of released cyanide in the Murchison meteorite suggested that this source of cyanide was separate from the cyanide responsible for the synthesis of some extraterrestrial amino acids and other organic compounds[13]. Furthermore, the compounds responsible for released cyanide in meteorites are still unknown.

Here, we report releasable cyanide abundance (often referred to as total cyanide) in acid-digested distillates of various meteorites by chemical derivatization (Supplementary Fig. 1) and liquid chromatography with fluorescence detection and time-of-flight mass spectrometry (LC-FD/ToF-MS). We identify two iron cyanocarbonyl complexes, $[Fe^{II}(CN)_5(CO)]^{3-}$ and $[Fe^{II}(CN)_4(CO)_2]^{2-}$, in the Lewis Cliff 85311 meteorite by liquid chromatography-high resolution orbitrap mass spectrometry. These extraterrestrial organometallic compounds are a source of free cyanide (HCN/CN$^-$) and also bear a striking similarity to portions of the active sites of [NiFe]- and [FeFe]-hydrogeneses, which suggests that these compounds may have played an important role during the origin and early evolution of life on Earth.

## Results

**Abundance and species of cyanide in meteorites.** We analyzed Allan Hills (ALH) 83100, Allan Hills 84001, Graves Nunataks (GRA) 06100, Lewis Cliff (LEW) 85311, Lewis Cliff 90500, Lonewolf Nunataks (LON) 94102, Murchison, and Roberts Massif (RBT) 04133 for acid-releasable cyanide. All of these meteorites are carbonaceous chondrites with the exception of ALH 84001, which is a martian meteorite (see Table 1 for classifications and additional information[14–16]). Figure 1 shows the unambiguous identification of cyanide in LEW 90500, a CM2 chondrite. There is a single peak at ~5.6 min. in the fluorescence chromatogram in addition to a single peak at ~5.7 min. in the extracted ion chromatogram (slight delay in retention time due to the mass spectrometer coming after the fluorescence detector), which is an exact match to our cyanide standard (in the form of an NDA-cyanide derivative). We also measured the same

fluorescence and mass peaks corresponding to cyanide for the other CM chondrites analyzed. All five of the CM chondrites we analyzed contain cyanide with LEW 85311 containing the highest concentration at 2472 ± 38 nmol CN·g$^{-1}$ meteorite (Table 1). The cyanide abundance for Murchison in our study (95 ± 1 nmol CN·g$^{-1}$ meteorite) was noticeably lower than the abundance (~400 nmol CN·g$^{-1}$ meteorite) previously reported by Pizzarello, although our extraction and analysis methods differed[12]. Method blanks were almost completely absent of cyanide resulting in a clean baseline (see Fig. 1 and Supplementary Note 1). We did not detect any (<0.1 nmol·g$^{-1}$) releasable cyanide in the CR2 (heated), CV3, or martian meteorite (see Supplementary Note 2). Cyanide can be found in nature, but most terrestrial sources are anthropogenic and are derived from industrial processes[17]. The observation that cyanide was group-specific (CMs only) among the Antarctic meteorites analyzed suggests that its source was indigenous to the meteorite rather than a pervasive terrestrial contaminant. Furthermore, Antarctic meteorites tend to be less contaminated compared to meteorites collected elsewhere[18–20] and cyanide abundances did not track with the meteorite's weathering grade, i.e., meteorites that have experienced greater terrestrial weathering effects did not have greater cyanide abundances.

All carbonaceous chondrites in the CM group have experienced some degree of pre-terrestrial aqueous alteration[21], which has been attributed to melting of ice inside asteroids from short-lived radionuclides ($^{26}$Al), electromagnetic induction, and/or impact heating[22]. During this aqueous alteration stage, fluids in the asteroid were likely low to moderate in temperature (0–150 °C) and alkaline in pH based on mineralogy and oxygen isotope data[23–26]. The prior detection of releasable cyanide in the water-soluble portion of the Murchison meteorite[12,13] suggested that this cyanide was not trapped in a mineral phase and would have been accessible to aqueous fluids in the meteorite parent body. Table 1 also shows CM chondrites associated with a numerical scale that estimates the degree of aqueous alteration (lower numbers mean more aqueously altered) from Alexander et al.[16]. There is a noticeable decrease in cyanide abundance for more aqueously altered CM chondrites. This trend suggests that the protracted aqueous alteration stage in the meteorite parent body may have altered or destroyed the compounds responsible for this cyanide, similar to previous studies involving the abundance of meteoritic amino acids and N-heterocycles[7,20,27].

Meteoritic insoluble organic matter (IOM) is a potential source of cyanide in CM chondrites because it represents the majority of organic matter in meteorites and is known to contain nitriles[28]. However, we measured a similar concentration of released cyanide in the Murchison meteorite to those measured in

### Table 1 Summary of cyanide abundances in meteorites

| Meteorite | Type | C (wt. %) | N (wt. %) | CN abundance (nmol·g$^{-1}$ meteorite)$^d$ | Aqueous alteration scale for CMs |
|---|---|---|---|---|---|
| ALH 83100 | CM1/2 | 1.90$^a$ | 0.070$^a$ | 50 ± 1 | 1.1$^c$ |
| Murchison | CM2 | 2.08$^a$ | 0.105$^a$ | 95 ± 1 | 1.6$^c$ |
| LEW 90500 | CM2 | 1.84 ± 0.04$^a$ | 0.094 ± 0.004$^a$ | 148 ± 6 | 1.6$^c$ |
| LON 94102 | CM2 | 2.06 ± 0.05$^a$ | 0.123 ± 0.003$^a$ | 421 ± 26 | 1.8$^c$ |
| LEW 85311 | CM2 | 2.03$^a$ | 0.156$^a$ | 2472 ± 38 | 1.9$^c$ |
| RBT 04133 | CV3 (reduced) | 0.06$^b$ | | <0.1 | |
| GRA 06100 | CR2 (heated) | 0.20 ± 0.01$^c$ | 0.010 ± 0.001$^c$ | <0.1 | |
| ALH 84001 | orthopyroxenite (martian) | | | <0.1 | |

$^a$From ref. 14
$^b$From ref. 15
$^c$From ref. 16
$^d$The error was calculated as the standard error of the mean from four measurements using mass and fluorescence data.
Source data are provided as a Source Data file

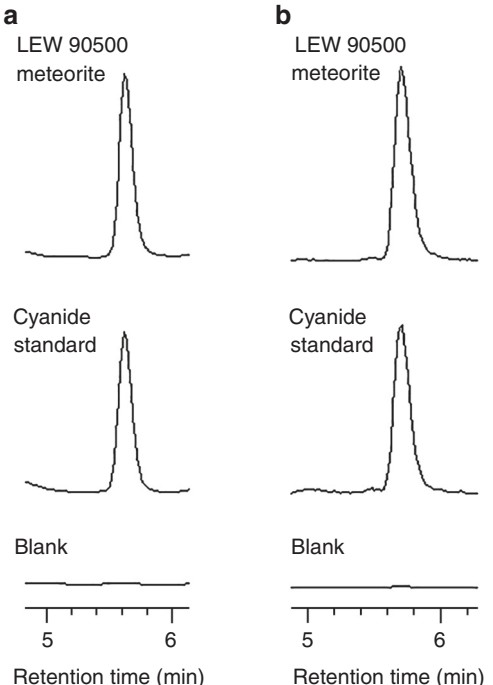

**Fig. 1** Identification of cyanide released from CM chondrites. **a** Fluorescence chromatograms ($\lambda_{ex}$ 252 nm, $\lambda_{em}$ 483 nm) of naphthalene-2,3-dicarboxaldehyde-cyanide derivative from the LEW 90500 meteorite, KCN standard, and method blank. **b** Extracted ion chromatograms ($m/z$ 251.08 with a ±0.03 window; $[M + H]^+$) of naphthalene-2,3-dicarboxaldehyde-cyanide derivative from the LEW 90500 meteorite, KCN standard, and method blank

previous studies of hot water extracts[12,13], the latter result indicates that the cyanide species would be water-soluble, which would rule out IOM as a source. If there was a high concentration of cyanide in meteorite parent bodies, cyanide may be present in polymeric form (we do not consider HCN polymers to be part of the IOM because they can be extracted in water yielding a yellow–brown solution[29]). However, it is unlikely that the cyanide detected in this study originated mainly from HCN polymer because the total cyanide liberated from the HCN polymer using our methods is only ~0.02% (wt/wt). The meteorite sample would need to be almost entirely HCN polymer in order to release the levels of cyanide we detect in LEW 85311 (see Supplementary Note 3). Organonitrile compounds do not appear to be a significant source of releasable cyanide in CM chondrites because they release little to no detectable cyanide when analyzed using our method (see Supplementary Note 4).

It is widely believed that many of the soluble organic compounds in carbonaceous chondrites were synthesized during the aqueous alteration stage(s) that occurred in the parent body, which also resulted in changes to the mineral composition[21]. Reaction-path calculations simulating aqueous alteration in CM parent bodies suggest that metal-organic compounds (along with carboxylic acids and amides, which agrees with experimental measurements of CM chondrites[3]) should dominate the composition of organic compounds[30]. Recently, dihydroxymagnesium carboxylates $[(OH)_2MgO_2C-R]^-$ were reported in the soluble organic fraction of various meteorites[31].

We analyzed both water and base extracts of LEW 85311 by inductively coupled plasma mass spectrometry in order to help elucidate potential speciation of cyanide in these meteorites. Both types of extractions had relatively similar element contents with high concentrations of Mg, Na, Ca, K, and Al. Appreciable

amounts of Fe, Ni, Ag, and Pt were also detected (Supplementary Table 1). Many of these elements are known to form simple cyanide salts such as NaCN and KCN; however, simple cyanides are only a minor contributor to the overall acid-releasable cyanide based on the analysis of LEW 90500 (see Supplementary Note 5). Iron forms particularly stable complexes with cyanide ligands due to cyanide's strong σ-donor properties. Using negative-ion electrospray ionization (ESI) mass spectrometry, we observed changes in iron oxidation state, loss of cyanide ligands, and aggregation of counterions to reduce charge in ferricyanide $[Fe^{III}(CN)_6]^{3-}$ and ferrocyanide $[Fe^{II}(CN)_6]^{4-}$ reference standards, which agrees with previous measurements of these complexes under ESI conditions[32]. A major identifying fragment for iron cyanide complexes is $[Fe^{II}(CN)_3]^-$ at $m/z$ 133.9447.

We identified this $[Fe^{II}(CN)_3]^-$ anion in high resolution mass spectra of base extracts of LEW 85311 meteorite based on accurate mass measurements and its distinct isotope pattern including a triplet peak with $^{13}C$, $^{15}N$, and $^{57}Fe$ isotopologues (Fig. 2). This mass fragment was not associated with either ferrocyanide or ferricyanide, rather it resulted from the fragmentation of iron cyanocarbonyl complexes $[Fe^{II}(CN)_5(CO)]^{3-}$ and $[Fe^{II}(CN)_4(CO)_2]^{2-}$ (Supplementary Fig. 2). $[Fe^{II}(CN)_3]^-$ is also a known fragment ion for both $[Fe^{II}(CN)_5(CO)]^{3-}$ and $[Fe^{II}(CN)_4(CO)_2]^{2-}$ in the negative-ion ESI mass spectrum[33]. We identified the parent ions $[H_2Fe^{II}(CN)_5(CO)]^-$ and $[HFe^{II}(CN)_4(CO)_2]^-$ using accurate mass measurements, their distinctive isotope patterns, and the observation that a single peak for each complex was present in their respective extracted ion chromatogram (Fig. 2 and Supplementary Fig. 2). We estimate that ~70% of the released cyanide was derived from these two organometallic compounds assuming similar responses of $m/z$ 133.9447 to ferrocyanide (see Supplementary Note 6). It is also noteworthy to point out that cyanide ligands stabilize CO binding to iron, which makes these iron cyanocarbonyl complexes very stable.

$[Fe^{II}(CN)_5(CO)]^{3-}$ and $[Fe^{II}(CN)_4(CO)_2]^{2-}$ (both *cis* and *trans* forms) have been synthesized in the laboratory from solutions containing ferrous salts ($FeCl_2$), $CN^-$, and $CO$[33], and similar synthetic routes were likely available during the aqueous alteration stage(s) on asteroids[30]. Additionally, the discovery of iron cyanocarbonyl complexes in aqueously altered meteorites suggests that these organometallic compounds could have also been produced by geochemical processes on early Earth[34–37]. Thus, we emphasize that the study of meteorites may help identify unusual or overlooked geochemical reactions (particularly those that are ancient or rarely seen today), as well as reveal new prebiotic geochemical compounds that were not actively being considered in scenarios relevant to the origin of life.

## Discussion

Both exogeneous and endogenous iron cyanocarbonyl complexes may have been important for prebiotic chemistry on early Earth. In the case of meteorites, aqueous leaching or dissolution of CM chondrites followed by UV irradiation of iron cyanocarbonyl complexes would have led to the release of cyanide from their metal centers[38]. Shallow ponds, streams, or puddles may have been favorable environments for cyanide release because they would have allowed greater UV penetration depth with less dilution effects. Some of this cyanide may have escaped into the atmosphere, particularly if the solution was acidic. Thus, exogenous cyanide delivery via meteorites may have provided a boost to the cyanide abundance produced on early Earth[34]; see Supplementary Discussion on estimated cyanide abundance on early Earth. Reactions involving cyanide were likely critical for the synthesis of numerous biologically relevant compounds and were

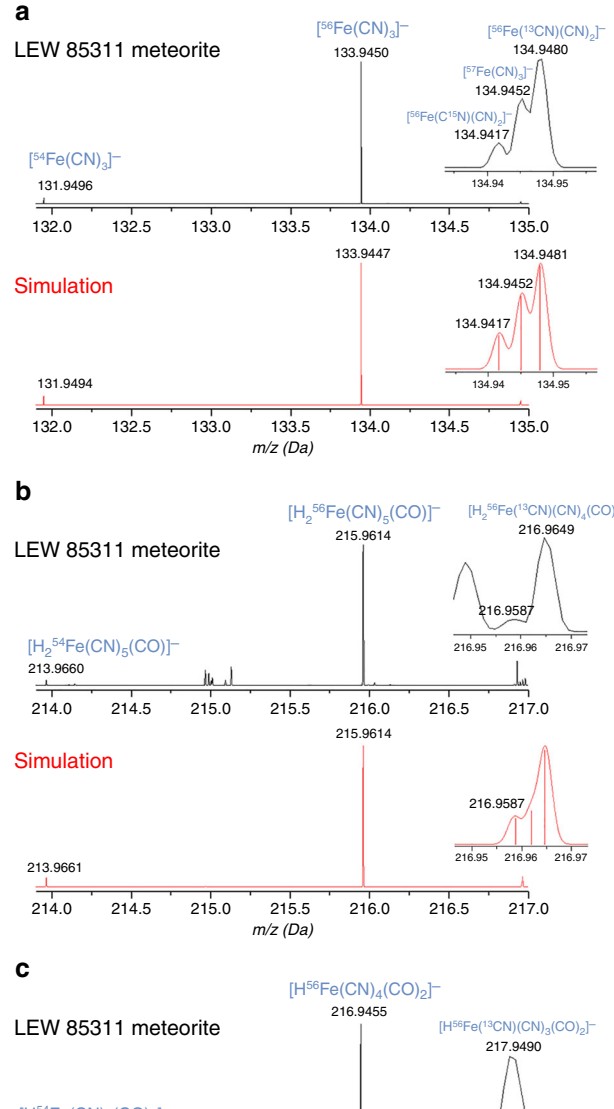

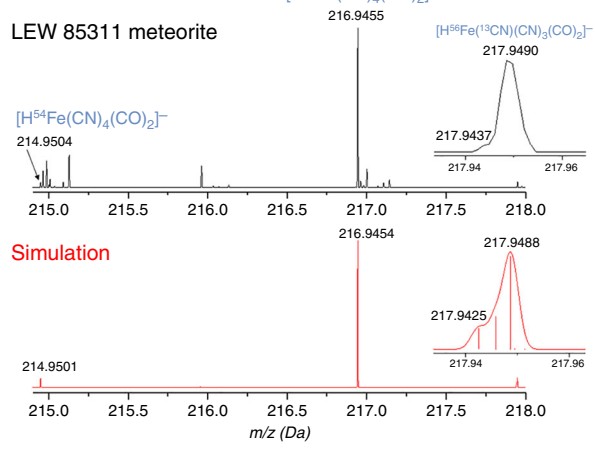

**Fig. 2** High resolution ESI mass spectra. **a** Mass spectrum of LEW 85311 meteorite extract along with simulated isotope pattern of $[Fe^{II}(CN)_3]^-$. **b** Mass spectrum of LEW 85311 meteorite extract along with simulated isotope pattern of $[H_2Fe^{II}(CN)_5(CO)]^-$. **c** Mass spectrum of LEW 85311 meteorite extract along with simulated isotope pattern of $[HFe^{II}(CN)_4(CO)_2]^-$. Accurate mass measurements and isotope patterns support the identification of two iron cyanocarbonyl complexes and a shared fragment ion. $H^+$ serves as counterions which reduce the overall charge of these species to −1. Simulated isotope patterns (in red) were generated using a Gaussian profile and a mass resolution of 65,000 resolution (full-width at half-maximum) in the Thermo Scientific XCalibur software

driven, in part, by photochemistry of organometallic compounds. For example, photochemical reduction of cyanohydrins by UV irradiation of iron cyanocarbonyl complexes in the presence of sulfite would have likely produced simple sugars and amino acid precursors since this process was previously demonstrated with ferrocyanide[39,40] (and we would expect similar photochemical behavior).

On early Earth, molecular hydrogen ($H_2$) was one of the earliest energy sources available and the ubiquity of hydrogen metabolism and deeply rooted lineages in both archaea and bacteria has led to the belief that hydrogen metabolism is ancient in origin[41]. Hydrogenases catalyze the reversible oxidation of molecular hydrogen ($H_2 \rightleftharpoons 2 H^+ + 2e^-$), which takes place at their organometallic active sites. In [NiFe]-hydrogenases as well as the phylogenetically unrelated [FeFe]-hydrogenases, the active sites have iron atoms coordinated to CO and CN ligands, which are assumed to be essential for enzyme activity because they are always present[42]. These CO and CN ligands are also considered unusual[43] since they are not found in other metalloenzymes. Figure 3 shows that the two iron cyanocarbonyl complexes found in LEW 85311 share structural similarities with the active sites of [NiFe]-hydrogenase from *Desulfovibrio gigas*[43,44] and [FeFe]-hydrogenase from *Clostridium pasteurianum*[45], specifically the $Fe^{II}(CN)(CO)$ moieties with similar geometries[33,46]. It is conceivable that iron cyanocarbonyl complexes may have served as the building blocks of these bimetallic active sites through dimerization (for [FeFe]-hydrogenases) or association with a nickel-containing species (for [NiFe]-hydrogenases) in conjunction with the loss of CN ligands (possibly through photodissociation).

A large number of model complexes mimicking the active site of [FeFe]-hydrogenases have been synthesized in the laboratory, although this number is much less for model complexes of [NiFe]-hydrogenase active sites due to the difficulty of synthesizing the heterobimetallic site[47]. Nevertheless, there have been reports of synthesizing model complexes of [NiFe]-hydrogenase active sites using Fe(CN)(CO)-containing precursors. For example, Perotto et al. demonstrated that stable heterobimetallic analogs of the active sites of [NiFe]-hydrogenases can be readily synthesized from reactions of *fac*-$[Fe(CO)_3(CN)_2I]$ and Ni-centered compounds containing polydentate ligands that were stirred for 1–12 h in acetonitrile[48]. Biomimetic complexes that are catalytically active have been synthesized in the laboratory[47], which suggests the possibility that primitive organometallic compounds could play a role in catalysis and hydrogen metabolism without a protein scaffold on early Earth.

## Methods

**Meteorite sample preparation**. All glassware and ceramics were rinsed with ultrapure water, wrapped in aluminum foil, and heated to 500 °C overnight in order to remove organic compounds. All meteorites were pulverized to a fine powder using a porcelain mortar and pestle under a positive pressure ISO 5 High-Efficiency Particulate Air (HEPA) filtered laminar flow hood (Labconco).

We acid-digested and distilled the following amounts of each meteorite: 204.3 mg ALH 83100 (specific 272, parent 33), 203.6 mg ALH 84100 (specific 425, parent 15), 206.7 mg GRA 06100 (specific 49, parent 47), 202.8 mg LEW 85311 (specific 78, parent 3), 209.3 mg LEW 90500 (specific 79, parent 2), 201.1 mg LON 94102 (specific 29, parent 15), 204.4 mg Murchison (USNM 5451.1), and 207.7 mg RBT 04133 (specific 21, parent 0). Each powdered meteorite sample was placed into a MicroDIST sample tube of the distillation apparatus (Lachat Instruments) along with 5.8 mL ultrapure water. 750 μL of 9 M $H_2SO_4$ was added to the sample tube as a releasing agent for cyanide and 1.5 mL of 0.08 M NaOH was added to the top of the tube as a trapping solution for cyanide. The sample tube was separated from the NaOH trapping solution by a gas-permeable membrane within the tube. The distillation tube was sealed with a press (Lachat Instruments) and placed into a hot oil bath with a temperature range of 134–142 °C for 30 min. The final distillate volume in the top tube after distillation was ~2–3 mL.

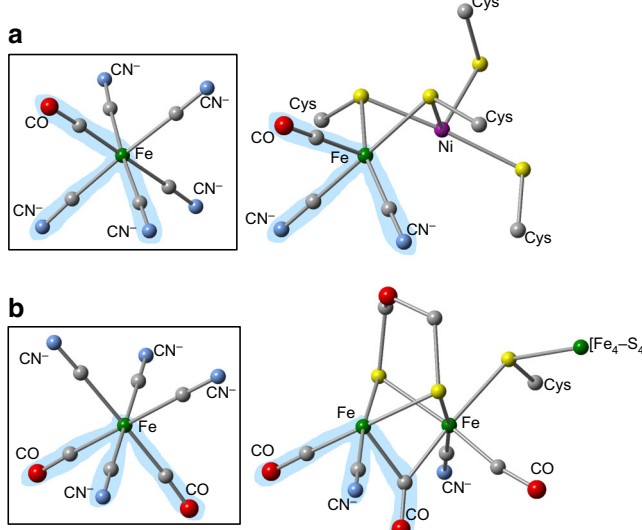

**Fig. 3** Meteoritic organometallic compounds compared to active sites. **a** $[Fe^{II}(CN)_5(CO)]^{3-}$ (boxed) and active-site structure of [NiFe]-hydrogenase from *Desulfovibrio gigas* (1FRV). **b** $[Fe^{II}(CN)_4(CO)_2]^{2-}$ (boxed, *cis* form shown) and active-site structure of [FeFe]-hydrogenase from *Clostridium pasteurianum* (3C8Y). Regions that are shaded blue indicate structural similarity. A bridging ligand between metals is not shown for clarity

Control samples were run in parallel with the meteorites, which included procedural (reagent) blanks, a serpentine blank (a hydrated magnesium silicate), and KCN standards.

**Derivatization**. Supplementary Figure 1 shows the derivatization scheme for cyanide using naphthalene-2,3-dicarboxaldehyde (NDA) and glycine. The meteorite and control sample distillates were processed using the following optimized conditions: derivatization for 15 min at room temperature using a mixture of 50 μL 0.1 M glycine, 50 μL 10 mM sodium borate buffer (pH 9.1), 50 μL meteorite extract or control sample, and 50 μL 1 mM NDA (modified from prior studies[49,50]). Glycine and sodium borate buffer solutions were dissolved in ultrapure water, KCN standards were dissolved in 0.08 M NaOH, and NDA was dissolved in methanol. Glycine, KCN, and NDA standard solutions were made fresh daily from the powder. We determined that the NDA-cyanide derivative showed <1% degradation after 2.25 h and <15% degradation after 17.5 h at room temperature. Due to the stability of the derivative, we derivatized and immediately analyzed one sample at a time to result in optimal response.

**Total cyanide analysis**. We used a Waters Acquity ultra performance liquid chromatograph coupled to a Waters Acquity fluorescence detector and Waters LCT Premier time-of-flight mass spectrometer for total cyanide analysis. Chromatographic separation was achieved using a prefilter and guard column followed by an Acquity BEH C18 2.1 × 50 mm column (1.7 μm particle size) and an Acquity BEH Phenyl 2.1 × 150 mm column (1.7 μm particle size) with a 25 μL injection volume. Samples were eluted using an isocratic flow of 35% 50 mM ammonium formate buffer (pH 8) with 8% methanol and 65% methanol at a flow rate of 150 μL min$^{-1}$ and column temperature of 30 °C. The fluorescence excitation wavelength for the NDA-cyanide derivative was 252 nm and the emission was monitored at 483 nm. MS settings were as follows: positive ion mode, ESI capillary voltage +3.5 kV, cone voltage 30 V, desolvation temperature 350 °C, and source 120 °C. The $N_2$ cone and desolvation gases were flowed at 50 L hr$^{-1}$ and 650 L hr$^{-1}$, respectively. Mass spectra were acquired in V-mode (mass resolution ~5000) with a range of $m/z$ 50–500. Cyanide identification in the meteorite was made by comparison of the fluorescence and single ion mass peak ($m/z$ 251.08 ± 0.03, which corresponds to the protonated molecular ion) retention times to standards. Cyanide concentration was calculated by comparing the fluorescence and mass peak areas of the sample to our standard concentration curve.

**Measurement of cyanide in standard solutions**. We performed a seven-point linearity study using non-distilled solutions of KCN with a concentration range of 0.01–2 μM (or 60 fmol–12 pmol on column). The curve was highly linear for fluorescence ($R^2 = 0.9997$) and for single ion mass detection at the theoretical protonated mass of 251.08 ± 0.03 Da ($R^2 = 0.9989$) (Supplementary Fig. 3). The limit of detection was 0.1 nmol CN·g$^{-1}$ meteorite (60 fmol on column) with a signal-to-noise-ratio of 3:1. The 0.5 μM KCN standard was processed through the entire

method from acid-digestion and distillation to derivatization and analyzed using LC-FD/ToF-MS in order to determine recovery. The average recovery for cyanide was 99% based on eight measurements with mass and fluorescence detection.

**Inductively coupled plasma mass spectrometry**. ICP-MS analysis was performed in medium resolution ($m/\Delta m = 4000$ at 1% peak height) using a Thermo Finnigan Element 2 ICP-MS in the Department of Geology Plasma Laboratory at the University of Maryland. All samples, blanks, and standards were aspirated via cyclonic nebulizer into the plasma source in 2% $HNO_3$ using Ar and $N_2$ carrier gases with a flow rate of 100 μL min$^{-1}$. The source was flushed extensively with ultrapure water between samples. The isotopes measured were: $^{23}Na$, $^{25}Mg$, $^{27}Al$, $^{39}K$, $^{43}Ca$, $^{47}Ti$, $^{49}Ti$, $^{56}Fe$, $^{57}Fe$, $^{59}Co$, $^{60}Ni$, $^{62}Ni$, $^{63}Cu$, $^{65}Cu$, $^{66}Zn$, $^{67}Zn$, $^{68}Zn$, $^{107}Ag$, $^{109}Ag$, $^{111}Cd$, $^{112}Cd$, $^{114}Cd$, $^{115}In$, $^{194}Pt$, $^{195}Pt$, $^{196}Pt$, and $^{197}Au$. Multiple mass stations were monitored for elements most susceptible to isobaric interferences (even in medium resolution), and $^{115}In$ spiked into each sample was used to track instrumental drift. Supplementary Table 1 shows the summary of elements measured in hot water and hot base extracts of LEW 85311.

**High resolution mass spectrometry**. Two samples of LEW 85311 (309.1 and 337.4 mg) were extracted under the following procedure: 500 μL of 0.08 M NaOH was added to each powdered LEW 85311 sample in 2.5-mL glass ampoules, flame-sealed, and put into an oven at 80 °C for ~20 h. A 0.08 M NaOH method blank was run in parallel with meteorite extracts.

Sample extracts were analyzed using a Thermo Scientific Accela High Speed LC coupled to a Thermo Scientific LTQ Orbitrap XL hybrid mass spectrometer. Separation was accomplished by injecting 20 μL sample solution onto a SIELC Primesep B2, 2.1 × 150 mm column (5 μm particle size). Mobile phase (A) was 100 mM ammonium acetate, pH = 4, mobile phase (B) was 100 mM ammonium acetate, pH = 9, mobile phase (C) was ultrapure water, and mobile phase (D) was acetonitrile. The elution method was as follows: initial mobile phase composition was set to 10% A, 70% C, and 20% D and held for 5 min, linear gradient to 50% B, 30% C, and 20% D over 15 min, mobile phase composition was returned to the initial conditions at 15.1 min, and the column was re-equilibrated for 15 min to finish the run. For optimal separation, the flow rate was set at 200 μL min$^{-1}$.

The Thermo Scientific LTQ Orbitrap XL hybrid mass spectrometer was equipped with an electrospray ionization (ESI) source and operated in negative-ion mode. Parameters for ESI were as follows: the nitrogen gas for desolvation of the electrospray was set to 40 for the sheath gas, 5 for the auxiliary gas, and 1 for the sweep gas (all in arbitrary units). The ion transfer capillary voltage and temperature were −21 V and 275 °C, respectively. Full scan spectra in negative-ion mode were acquired over a mass range of $m/z$ 120–320 and automated gain control (AGC) was set to $5 \times 10^5$ ions with a maximum injection time of 400 millisecond. The mass resolution was set to 30,000 (which translates to a mass resolution closer to 60,000 at $m/z$ 200) to obtain good chromatographic peaks (~14 mass spectra were collected over a chromatographic peak width of 10 s). External calibration was performed using a mixture of sodium dodecyl sulfate, sodium taurocholate, and Ultramark 1621 in an acetonitrile-methanol-water solution containing 1% acetic acid.

Elemental compositions were calculated from the negative ion using C, H, O, N, Na, K, and Fe. Accurate mass measurements (usually <1 ppm mass error) enabled assignment of molecular formulas, which were consistent with predicted isotope patterns. While the orbitrap mass analyzer is known for its high mass accuracy and resolving power, its spectral accuracy[51] (i.e., accuracy in measuring the abundances of isotopic peaks) has been known to underrepresent heavier isotopologues[52], which may be the case for a couple of low intensity isotopic peaks belonging to iron cyanocarbonyl complexes. Nevertheless, accurate mass and distinct isotope patterns with good spectral accuracy enabled confident molecular assignments in our study.

## Data availability
Data from this manuscript will be made available on Boise State University ScholarWorks (https://scholarworks.boisestate.edu/). This includes raw data as well as figures for both the main manuscript and the Supplementary Information. Alternatively, raw data and figure image files are available on request from the corresponding author M.P.C. The source data underlying Table 1, Supplementary Fig. 3, and Supplementary Table 1 are provided as a Source Data file.

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

## Acknowledgements

Support for this project was provided by the NASA Emerging Worlds Program (11-COS11-0061 and NNX16AP59G) and NASA Astrobiology Institute via the Penn State Astrobiology Research Center (cooperative agreement #NNA09DA76A) and the Goddard Center for Astrobiology (CAN 5). K.E.S. also acknowledges support from the NASA Earth and Space Science Fellowship and the NASA Pennsylvania Space Grant Consortium. We thank Prof. Eric Brown (Boise State University) for discussions and assistance with making one of the figures and the Plasma Mass Spectrometry Laboratory in the Department of Geology at the University of Maryland for use of their ICP-MS instrument. US Antarctic meteorite samples are recovered by the Antarctic Search for Meteorites (ANSMET) program, which has been funded by NSF and NASA, and characterized and curated by the Department of Mineral Sciences of the Smithsonian Institution and Astromaterials Acquisition and Curation Office at NASA Johnson Space Center.

## Author contributions

K.E.S. and M.P.C. designed research with input from C.H.H., R.D.A. and J.P.D.; K.E.S., R.D.A. and M.P.C. performed research; C.H.H., J.P.D. and M.P.C. contributed new reagents/analytic tools; K.E.S., R.D.A. and M.P.C. analyzed data; K.E.S. and M.P.C. wrote the paper with contributions from C.H.H., R.D.A. and J.P.D.

## Additional information

**Competing interests:** The authors declare no competing interests.

