## [Peer Review File · Nature Communications]

Reviewers' comments:

Reviewer #1 (Remarks to the Author):

Review of "Organometallic compounds as carriers of extraterrestrial cyanide in primitive meteorites" by Smith et al.

This paper contains an excellent work to find iron cyanocarbonyl complexes in a carbonaceous meteorite, which might be a main reservoir of cyanide in meteorites.

The cyanide (CN⁻) is a very important material to promote chemical evolution in the universe. This paper should be published in Nature Communications. Before its acceptance, I recommend a minor revision for clarification and corrections.

< Main comments >

1) A major concern is the concentration of iron cyanocarbonyl complexes in the meteorite. While [Fe(II)(CN)₅(CO)]³⁻ and [Fe(II)(CN)₄(CO)₂]²⁻ were identified in the LEW 85311 extract by high-resolution mass spectrometry (Fig. 2), there was no description about its concentration. Assuming that ~70% the released cyanide (~1.7 μmol/g from Table 1) was derived from the two iron cyanocarbonyl complexes (lines 124-125), the concentration of iron cyanocarbonyl complexes is estimated to be ~0.4 μmol/g (4-5 CN per Fe-complex). Is this correct? Please describe how to estimate the concentration of the Fe-complexes using the intensity of m/z 133.9447 to ferrocyanide (line 125). And if so, the extractable Fe (0.44 μmol/g for the base extract, Table S1) occurs mostly as the cyanocarbonyl complexes. It may be interesting to note that the CN content is an important factor to control the concentration of the extractable Fe.

2) Another concern is why the cyanide content negatively correlates with the degree of aqueous alteration. Please describe the mechanism(s) to decompose (or not synthesize) the iron cyanocarbonyl complexes in more altered meteorites. At line 74-75, "The compounds responsible for this cyanide likely were synthesized during the protracted aqueous alteration stage" is not consistent with the above observation? Please clarify the effect of aqueous alteration in more detail.

3) I am not sure of the relationship between the Fe cyanocarbonyl complex and active site of NiFe and FeFe hydrogenase in Fig. 3. The binuclear complexes may be more difficult to form in natural environments. Please provide any data to support the relationship including thermodynamic data for the stability of both complexes and/or formation mechanism of binuclear complexes from the Fe cyanocarbonyl complex observed in this study. Incidentally, Ni-cyanocarbonyl complexes have been searched in the extract?

< Other minor comments >

L111: ferrocyanide is [Fe(II)(CN)₆]⁴⁻, and ferricyanide is [Fe(III)(CN)₆]³⁻.

L144-158: Photochemistry ("UV irradiation" or "UV penetration") is mandatory for chemical reactions of Fe cyanocarbonyl complexes? The thermochemical reaction of Fe cyanocarbonyl complexes is not possible inside asteroids or early Earth?

L215-224 or Fig. 1: Please describe chromatographic conditions for LC-FD/ToF-MS (e.g. separation column, eluent(s), flow rate, etc).

L249-261 or Fig. S2: Again describe chromatographic conditions for high resolution MS (e.g. separation column, eluent(s), flow rate, etc).

Table 1: Please give carbon (and possibly nitrogen) content for each meteorite, if known.

Figure 2: Does the isotope simulation use terrestrial isotope composition of each element? If so, the identified Fe-complexes in the meteorite did not have an extreme different isotopic composition from terrestrial compositions?

Reviewer #2 (Remarks to the Author):

Dear Editors, dear Authors,

the manuscript "Organometallic compounds as carriers of extraterrestrial cyanide in primitive meteorites" by Callahan et al. seems very interesting to me. Its contents is of high interest for a broad scientific community, including the research fields of astrochemistry, astrobiology and meteoritics. This work is novel and discusses an important issue, the interaction between organometallic and organic astrochemical compounds. The papers results will potentially influence the research field on the origin(s) of life. The manuscript is well-written, in clear and exciting manner.

My review comments are placed as notes within the pdf file '192590_0_art_file_3456217_pjr333_convrt reviewer.pdf'. I kindly ask the authors to refine the manuscript according to my comments. In its present form, I cannot fully recommend the paper for publication in Nature Communications. Next to some technical comments and general suggestions, I ask the authors to mainly refine the manuscript according two major issues:

- (i) evidence for nonterrestrial origin
- (ii) speculative statements

(i) What gives evidence for nonterrestrial origin? How is it ruled out that the measured organometallics is no terrestrial contamination? In astrochemical research, special care on terrestrial contamination is a key aspect. A weak/nonlinear relation to asteroidal/meteoritic aqueous alteration with only five samples is not a strong argument to rule out that the detected compounds are not of terrestrial origin. I would like to encourage the authors to re-discuss this point and to stronger verify the nonterrestrial origin of the observed iron cyanocarbonyl compounds.

(ii) Some statements are speculative (e.g. sample heterogeneity of Murchison as explanation for difference in detected CN^- concentrations; "The remaining 30% is likely due to simple cyanide salts and possibly other cyanometallates and is suggestive of a broader metal-organic chemistry preserved in meteorites."). Please stay on solid facts within the whole paper.

Reviewer #3 (Remarks to the Author):

The authors report cyanide abundances in meteorites and showed that the majority of cyanide is in the form of iron cyanocarbonyl complexes. The authors then discuss that these iron cyanocarbonyl complexes derived to the Earth by meteorites would have important roles in prebiotic chemistry. Although contributions of these compound to prebiotic chemistry have some uncertainty, I agree with the authors about the importance of the cyanide compounds in meteorites and I generally recommend for publication in Nature Communications. However, I have a concern about their results, that I would expect the authors can address before publication.

The major concern:

I suspect that the meteoritic cyanides really in the form of iron cyanocarbonyl complexes in meteorites, i.e., is there any possibility that the iron cyanocarbonyl complexes were derived from another form of cyanides reacted with Fe ions during analytical procedures such as extraction, derivatization and ionization (ESI)?

Minor issues:

L32-34: References #1 and #2 do not seem to be appropriate for such broad statements. I recommend to refer review papers.

L61: Please mention here that the CR2 is heated.

L65: Most people agree that ^{26}Al is major heat source but it is not the only one.

L74-75: "...the compounds responsible for this cyanide likely were synthesized during the protracted aqueous alteration stage"

I think this statement is inconsistent with the results showing that more-altered CMs have lower abundance of cyanides.

L92: aqueous alteration occurred "in" the parent body rather than "on" the parent body.

L181: Aluminum foil?

L258: Is the number of mass resolution setting 30,000 correct? Because it seems not very high. (I am not a specialist of Orbitrap, so just ignore this comment if it is correct number)

L 740-749: Please check the titles of references #42-45.

Reviewer #1 (Remarks to the Author):

Review of “Organometallic compounds as carriers of extraterrestrial cyanide in primitive meteorites” by Smith et al.

This paper contains an excellent work to find iron cyanocarbonyl complexes in a carbonaceous meteorite, which might be a main reservoir of cyanide in meteorites. The cyanide (CN⁻) is a very important material to promote chemical evolution in the universe. This paper should be published in Nature Communications. Before its acceptance, I recommend a minor revision for clarification and corrections.

We thank the Reviewer for his/her comments and the recommendation that our manuscript should be published in Nature Communications.

< Main comments >

1) A major concern is the concentration of iron cyanocarbonyl complexes in the meteorite. While $[\text{Fe}^{\text{II}}(\text{CN})_5(\text{CO})]^{3-}$ and $[\text{Fe}^{\text{II}}(\text{CN})_4(\text{CO})_2]^{2-}$ were identified in the LEW 85311 extract by high-resolution mass spectrometry (Fig. 2), there was no description about its concentration.

Assuming that ~70% the released cyanide (~1.7 $\mu\text{mol/g}$ from Table 1) was derived from the two iron cyanocarbonyl complexes (lines 124-125), the concentration of iron cyanocarbonyl complexes is estimated to be ~0.4 $\mu\text{mol/g}$ (4-5 CN per Fe-complex). Is this correct? Please describe how to estimate the concentration of the Fe-complexes using the intensity of m/z 133.9447 to ferrocyanide (line 125). And if so, the extractable Fe (0.44 $\mu\text{mol/g}$ for the base extract, Table S1) occurs mostly as the cyanocarbonyl complexes. It may be interesting to note that the CN content is an important factor to control the concentration of the extractable Fe.

That is absolutely correct! We have added a new section in Supplementary Information describing how we estimated the concentration of iron cyanocarbonyl complexes in LEW 85311 meteorite. Also, we completely agree with the Reviewer that it is very interesting that the CN content seems to be an important contributing factor towards the amount of solvent-extractable Fe. We make mention of this in our revised text as well, which is shown below for convenience.

Estimated concentration of iron cyanocarbonyl complexes (and its releasable cyanide) in LEW 85311. To estimate the abundance of iron cyanocarbonyl complexes in a 0.08 M NaOH extraction of LEW 85311, we used the peak areas from the two peak at 12 min. and 12.5 min. generated from the extracted ion chromatogram at m/z 133.9447 (refer to bottom trace of Fig. S2) and compared them to the peak area generated from the extracted ion chromatogram at m/z 133.9447 of a 100 μM ferrocyanide standard solution. The peak at m/z 133.9447 is a $[\text{Fe}(\text{CN})_3]^-$ fragment ion common to both iron cyanocarbonyl complexes as well as the ferrocyanide reference standard. We estimated the abundance of $[\text{Fe}^{\text{II}}(\text{CN})_5(\text{CO})]^{3-}$ and $[\text{Fe}^{\text{II}}(\text{CN})_4(\text{CO})_2]^{2-}$ at 266 and 98 $\text{nmol}\cdot\text{g}^{-1}$, respectively, using the assumption that the signal response at m/z 133.9447 for ferrocyanide standard is identical to the signal response at m/z 133.9447 for each iron cyanocarbonyl complex at the same concentration. We also note the observation that cyanide

content (est. $0.365 \mu\text{mol}\cdot\text{g}^{-1}$ in the form of $[\text{Fe}^{\text{II}}(\text{CN})_5(\text{CO})]^{3-}$ and $[\text{Fe}^{\text{II}}(\text{CN})_4(\text{CO})_2]^{2-}$) seems to be an important contributing factor towards the amount of solvent-extractable iron (ranging from $0.44 - 0.66 \mu\text{mol}\cdot\text{g}^{-1}$ based on the extraction type). Next, if we assume complete dissociation of cyanide ligands from their iron centers, the releasable cyanide from $[\text{Fe}^{\text{II}}(\text{CN})_5(\text{CO})]^{3-}$ and $[\text{Fe}^{\text{II}}(\text{CN})_4(\text{CO})_2]^{2-}$ is calculated at 1332 and 393 $\text{nmol}\cdot\text{g}^{-1}$, respectively, for a total releasable cyanide concentration of 1726 $\text{nmol}\cdot\text{g}^{-1}$, which is about 70% of the acid releasable cyanide in LEW 85311 (2472 $\text{nmol}\cdot\text{g}^{-1}$, which was reported in Table 1).

2) Another concern is why the cyanide content negatively correlates with the degree of aqueous alteration. Please describe the mechanism(s) to decompose (or not synthesize) the iron cyanocarbonyl complexes in more altered meteorites. At line 74-75, “The compounds responsible for this cyanide likely were synthesized during the protracted aqueous alteration stage” is not consistent with the above observation? Please clarify the effect of aqueous alteration in more detail.

We have modified the text so that our observations are consistent with our experimental data. The text (lines 83-87 now) reads: There is a noticeable decrease in cyanide abundance for more aqueously altered CM chondrites. This trend suggests that the protracted aqueous alteration stage in the meteorite parent body may have altered or destroyed the compounds responsible for this cyanide, similar to previous studies involving the abundance of meteoritic amino acids and N-heterocycles^{7, 20, 27}.

3) I am not sure of the relationship between the Fe cyanocarbonyl complex and active site of NiFe and FeFe hydrogenase in Fig. 3. The binuclear complexes may be more difficult to form in natural environments. Please provide any data to support the relationship including thermodynamic data for the stability of both complexes and/or formation mechanism of binuclear complexes from the Fe cyanocarbonyl complex observed in this study. Incidentally, Ni-cyanocarbonyl complexes have been searched in the extract?

To our knowledge, the synthesis of the binuclear complex (or close analog) under natural (prebiotic) conditions is relatively unexplored and there is no known experimental or computational data regarding the formation mechanism of the binuclear complex starting explicitly from $[\text{Fe}^{\text{II}}(\text{CN})_5(\text{CO})]^{3-}$ or $[\text{Fe}^{\text{II}}(\text{CN})_4(\text{CO})_2]^{2-}$, which were the complexes we measured in LEW 85311 meteorite. However, there have been reports using similar $\text{Fe}(\text{CN})(\text{CO})$ -containing precursors. For example, Perotto et al. (2018) demonstrated that stable heterobimetallic analogs of the active sites of [NiFe] hydrogenases can be *readily synthesized* from reactions containing $\text{fac-}[\text{Fe}(\text{CO})_3(\text{CN})_2\text{I}]$ with $[\text{Ni}(\text{L}^{1-3})]$ (L are different polydentate ligands). There are also numerous publications describing the synthesis of hydrogenase active site analogs that demonstrate that a homobimetallic or heterobimetallic center can readily form (albeit from precursors different from iron cyanocarbonyls and typical inorganic reaction conditions). We have added text (lines 183-191) to include these points in our manuscript.

Reaction schemes shown above are from Perotto et al., *Inorg. Chem.* 2018, 57, 2558–2569.

Finally, since our discovery of organometallic compounds in meteorites, we have been actively searching for more compounds (including nickel-containing species). You'll probably hear more about it from us in the future.

< Other minor comments >

L111: ferrocyanide is $[\text{Fe}^{\text{II}}(\text{CN})_6]^{4-}$, and ferricyanide is $[\text{Fe}^{\text{III}}(\text{CN})_6]^{3-}$.

Thank you for pointing this out. This mistake has now been corrected.

L144-158: Photochemistry (“UV irradiation” or “UV penetration”) is mandatory for chemical reactions of Fe cyanocarbonyl complexes? The thermochemical reaction of Fe cyanocarbonyl complexes is not possible inside asteroids or early Earth?

We are not sure whether we understand the Reviewer’s questions here. We wrote this section to propose one hypothetical scenario where iron cyanocarbonyl complexes in meteorites were leached on early Earth and then participated in subsequent reactions such as the release of cyanide or driving the synthesis of other complex organic compounds. However, photochemistry is *not* mandatory for chemical reactions involving iron cyanocarbonyl complexes. For example, we used heat and a strong acid to release cyanide from these complexes in our experiments with meteorites.

L215-224 or Fig. 1: Please describe chromatographic conditions for LC-FD/ToF-MS (e.g. separation column, eluent(s), flow rate, etc).

These instrumental details have now been included in our manuscript.

L249-261 or Fig. S2: Again describe chromatographic conditions for high resolution MS (e.g. separation column, eluent(s), flow rate, etc).

These instrumental details have now been included in our manuscript.

Table 1: Please give carbon (and possibly nitrogen) content for each meteorite, if known.

We modified Table 1 to include bulk carbon and nitrogen composition from previously published data that we could locate. It is interesting to note that bulk nitrogen composition seems to be increasing with less aqueously altered CMs, and these were the meteorites with the highest (acid-releasable) cyanide abundances.

Figure 2: Does the isotope simulation use terrestrial isotope composition of each element? If so, the identified Fe-complexes in the meteorite did not have an extreme different isotopic composition from terrestrial compositions?

The isotope simulation uses terrestrial isotope composition. The observation that identified iron cyanocarbonyl complexes in the meteorite did not have an extreme different isotopic composition from terrestrial compositions is to be expected. Please note that compound-specific isotope ratio mass spectrometry used to identify organic compounds in meteorites as terrestrial or extraterrestrial report values in units of per mil (‰, parts per thousand) because the magnitude of isotope fractionation is typically very small and such information cannot be distinguished by looking at peaks in the mass spectrum by eye. This extends to iron as well. For example, bulk Murchison (CM2 chondrite), like all chondrites, has a $\delta^{56}\text{Fe}$ and $\delta^{57}\text{Fe}$ of 0.00‰ within error (Hezel et al., 2018). Furthermore, spectral accuracy, which is the mass analyzer's ability to accurately measure isotopic distribution, on an Orbitrap can vary significantly (for example, see Erve et al., "Spectral Accuracy of Molecular Ions in an LTQ/Orbitrap Mass Spectrometer and Implications for Elemental Composition Determination"). We did not expect to distinguish between terrestrial or extraterrestrial isotope patterns for any compound in the mass spectrum; however, we did use isotope patterns to help confirm identity/molecular formulas.

Reviewer #2 (Remarks to the Author):

Dear Editors, dear Authors,

the manuscript "Organometallic compounds as carriers of extraterrestrial cyanide in primitive meteorites" by Callahan et al. seems very interesting to me. Its contents is of high interest for a broad scientific community, including the research fields of astrochemistry, astrobiology and meteoritics. This work is novel and discusses an important issue, the interaction between organometallic and organic astrochemical compounds. The papers results will potentially influence the research field on the origin(s) of life. The manuscript is well-written, in clear and exciting manner.

My review comments are placed as notes within the pdf file '192590_0_art_file_3456217_pjr333_convrt reviewer.pdf'. I kindly ask the authors to refine the manuscript according to my comments. In its present form, I cannot fully recommend the paper for publication in Nature Communications. Next to some technical comments and general suggestions, I ask the authors to mainly refine the manuscript according two major issues:

(i) evidence for nonterrestrial origin

(ii) speculative statements

(i) What gives evidence for nonterrestrial origin? How is it ruled out that the measured organometallics is no terrestrial contamination? In astrochemical research, special care on terrestrial contamination is a key aspect. A weak/nonlinear relation to asteroidal/meteoritic aqueous alteration with only five samples is not a strong argument to rule out that the detected compounds are not of terrestrial origin. I would like to encourage the authors to re-discuss this point and to stronger verify the nonterrestrial origin of the observed iron cyanocarbonyl compounds.

We thank the Reviewer for bringing this very important point to our attention. After looking again at our originally submitted manuscript, we should have done a better job stating our case why we believe both cyanide and iron cyanocarbonyl complexes were nonterrestrial in origin. We list the following reasons below.

1. Pizzarello showed that water-extractable, acid-releasable HCN from Murchison had $\delta^{13}\text{C}$ and $\delta^{15}\text{N}$ values consistent with an extraterrestrial origin. We make this clearer now on lines 36-37. We inferred that CN detected in Murchison in our study would also be extraterrestrial. We also surmised that CN released in CM2s would likely not be unique to just Murchison (which was confirmed by our experimental results).

2. “Cyanide can be found in nature, but most terrestrial sources are anthropogenic and are derived from industrial processes¹⁷. The observation that cyanide was group-specific (CMs only) among the Antarctic meteorites analyzed suggests that its source was indigenous to the meteorite rather than a pervasive terrestrial contaminant. Furthermore, Antarctic meteorites tend to be less contaminated compared to meteorites collected elsewhere¹⁸⁻²⁰ and cyanide abundances did not track with the meteorite’s weathering grade, i.e., meteorites that have experienced greater terrestrial weathering effects did not have greater cyanide abundances.” This text was added to our manuscript on lines 64-72.

3. “There is a noticeable decrease in cyanide abundance for more aqueously altered CM chondrites. This trend suggests that the protracted aqueous alteration stage in the meteorite parent body may have altered or destroyed the compounds responsible for this cyanide, similar to previous studies involving the abundance of meteoritic amino acids and N-heterocycles^{7, 20, 27}.” This text was added to our manuscript on lines 83-87.

4. With past meteorite studies, suspected terrestrial contamination was recognized from the presence of biologically common organic compounds (e.g., *n*-alkanes, L-amino acids with characteristic distributions and/or terrestrial isotopic ratios) or anthropogenic pollutants (PAHs/alkylated PAHs). To our knowledge, there are no reports of either $[\text{Fe}^{\text{II}}(\text{CN})_5(\text{CO})]^{3-}$ or $[\text{Fe}^{\text{II}}(\text{CN})_4(\text{CO})_2]^{2-}$ in biology. We had to go back a ways to find >100-year old reports of $[\text{Fe}^{\text{II}}(\text{CN})_5(\text{CO})]^{3-}$ salts in industrial use – for example, $[\text{Fe}^{\text{II}}(\text{CN})_5(\text{CO})]^{3-}$ is recovered during the purification of coal gas (H.E. Williams, 1915); there is no mention of $[\text{Fe}^{\text{II}}(\text{CN})_4(\text{CO})_2]^{2-}$. In 1887, Muller showed that a heating a solution of ferrocyanide in an atmosphere of CO for days converts a portion of ferrocyanide into carbonylferrocyanide. It’s worth mentioning that natural ferrocyanide is also rare (or at least rare in the scientific literature). For example, we could only

find one report - soluble ferrocyanide in the thermal springs of caldera have been reported by Mukhin 1974. We arrived at the conclusion that $[\text{Fe}^{\text{II}}(\text{CN})_5(\text{CO})]^{3-}$ and $[\text{Fe}^{\text{II}}(\text{CN})_4(\text{CO})_2]^{2-}$ in Antarctic meteorites are highly unlikely to be terrestrial contaminants.

(ii) Some statements are speculative (e.g. sample heterogeneity of Murchison as explanation for difference in detected CN^- concentrations; "The remaining 30% is likely due to simple cyanide salts and possibly other cyanometallates and is suggestive of a broader metal-organic chemistry preserved in meteorites."). Please stay on solid facts within the whole paper.

Thank you for bringing this to our attention. We have removed the speculative statements that were indicated by the Reviewer. We agree with the Reviewer that these statements were not necessary and their removal keeps the focus on our new and exciting results.

For example, statements about sample heterogeneity are commonly (and all too easily) invoked reasoning for differences in compound abundances in different samples of the same meteorite. However, there are notable differences in the meteorite extraction and analysis technique we developed and utilized compared to those of Pizzarello's study of Murchison. We make mention of this, which was part of our effort to increase the use of statements of facts and observations and reduce the number of speculative statements.

There are other comments in the annotated pdf provided by the reviewer and we have addressed many of these comments in our revised manuscript. Based on three Reviewers' comments, we have significantly revised our manuscript. We also have provided a source data file (per Nature Communications guidelines) and updated our data availability statement. We believe that our paper is much improved and ready for acceptance for publication. We thank the Reviewer for enabling these improvements.

Reviewer #3 (Remarks to the Author):

The authors report cyanide abundances in meteorites and showed that the majority of cyanide is in the form of iron cyanocarbonyl complexes. The authors then discuss that these iron cyanocarbonyl complexes derived to the Earth by meteorites would have important roles in prebiotic chemistry.

Although contributions of these compound to prebiotic chemistry have some uncertainty, I agree with the authors about the importance of the cyanide compounds in meteorites and I generally recommend for publication in Nature Communications. However, I have a concern about their results, that I would expect the authors can address before publication.

The major concern:

I suspect that the meteoritic cyanides really in the form of iron cyanocarbonyl complexes in meteorites, i.e., is there any possibility that the iron cyanocarbonyl complexes were derived from another form of cyanides reacted with Fe ions during analytical procedures such as extraction, derivatization and ionization (ESI)?

The Reviewer suggests that iron cyanocarbonyl complexes could be formed *in situ* during extraction, derivatization, or ionization. We recognize that past studies (not any of ours though!)

have reported meteoritic organic compounds could have been the result of the analytical procedure used; however, we are confident that this is **not** the case with our study and provide the following rationale.

With regards to derivatization, it was not used in the analytical procedure analyzing iron cyanocarbonyl complexes so it can be ruled out. Derivatization was only used for the analysis of total (acid releasable) cyanide.

Formation of these complexes during electrospray ionization is extremely unlikely, as complex formation would require that the individual components exhibit identical residence times on the HPLC column in order to be present at the time of ionization. Even if we take the highly improbable assumption that some unknown precursors could make it to the electrospray ionization step as true, there would be not enough cyanide ligands to contribute towards the formation of these complexes. We base this opinion on the free and total cyanide analysis of LEW 90500, which showed that very little of the total (acid releasable) cyanide was in the form of free cyanide (HCN, CN⁻) or simple cyanide salts (NaCN, KCN) that could act as a potential source of ligands.

Complex formation during extraction under alkaline conditions is also highly unlikely, as formation would require an improbable conjunction of events and conditions. For example, the iron(II) cyanocarbonyls in question have been deliberately synthesized (see Koch, 2001 & 2002) through the reaction of [Fe(CN)₆]⁴⁻ or [Fe(CN)₅(NH₃)]³⁻ with CO at elevated temperature and pressures, or alternatively *via* the reaction of Fe(CO)₄I₂ with CN⁻, also under forcing conditions. By contrast, not only were the extraction conditions described in this work milder than those reportedly required to form the cyanocarbonyl complexes, but the extractions were performed in the absence of the requisite CO or CN⁻ components. Furthermore, if the cyanocarbonyls were indeed products of the extraction process, one would expect to observe evidence of the parent precursor complexes (*e.g.* [Fe(CN)₆]⁴⁻). As stated in the manuscript, no evidence of such precursors was detected.

One might also speculate that the cyanocarbonyls could be derived from a low-valent iron carbonyl species present in the carbonaceous chondrite and somehow converted to the cyanide-bearing complex during alkaline extraction. Indeed, it is well-known that metal carbonyls can react with hydroxide ions to produce metal carbonylate anions, the archetypal example being the conversion of Fe(CO)₅ to the [HFe(CO)₄]⁻ anion under alkaline conditions. It is important to note, however, that an alkaline environment provides *reducing* conditions for metal carbonyls, and the iron cyanocarbonyls identified in the manuscript are encountered in an oxidized form (*i.e.* Fe²⁺). Thus, the following series of chemical events would be required to occur during the relatively short extraction process: 1) the reduction of Fe⁰ to Fe²⁻, 2) oxidation of Fe²⁻ to Fe²⁺, and 3) coordination of CN⁻ to the iron center. This orchestrated sequence of events is highly improbable, becoming even more so when one considers that there may have not been much free CN⁻ in meteorites. Even if free Fe(CO)₅ could be converted to [HFe(CO)₄]⁻ under the extraction conditions employed in this study, an important question could be raised - *Why would the metal carbonyl not have already undergone reduction in the alkaline environment of the asteroidal parent body during the protracted aqueous alteration stage(s)?*

Minor issues:

L32-34: References #1 and #2 do not seem to be appropriate for such broad statements. I recommend to refer review papers.

We agree and we have included many, more appropriate references including review papers.

L61: Please mention here that the CR2 is heated.

We have included this mention.

L65: Most people agree that ^{26}Al is major heat source but it is not the only one.

We have modified the text to include other means of heating that have been proposed such as electromagnetic induction and impact heating and included the appropriate reference.

L74-75: "...the compounds responsible for this cyanide likely were synthesized during the protracted aqueous alteration stage"

I think this statement is inconsistent with the results showing that more-altered CMs have lower abundance of cyanides.

This point was also brought up by another reviewer and we have revised the text.

L92: aqueous alteration occurred "in" the parent body rather than "on" the parent body.

We have revised the text and replaced "on" with "in". Actually, we caught this in a lot of sentences! So thank you for pointing this out.

L181: Aluminum foil?

We have it specified now.

L258: Is the number of mass resolution setting 30,000 correct? Because it seems not very high. (I am not a specialist of Orbitrap, so just ignore this comment if it is correct number)

The mass resolution ($m/\Delta m$) of the orbitrap mass analyzer is selected using predefined settings - for example, a mass resolution of 30,000 at m/z 400. It's important to point out that mass resolution here scales depending on the mass, i.e. at $m/z < 400$ the mass resolution increases from 30,000 and at $m/z > 400$ the mass resolution decreases from 30,000. In the case of iron cyanocarbonyl complexes and its distinct mass fragment, the mass resolution is closer to 60,000, which enables the isotope patterns to be revealed. We have added text to clarify this point.

We would also like to point out that mass resolutions of 30,000 and 60,000 are actually very high for LC-MS analyses (especially when increasing mass resolution also increases the MS acquisition time, which would affect the number of MS scans across a chromatographic peak).

To give you a comparison, the mass resolution of the LC-FD/ToF-MS method was only 5,000, but this resolution was perfectly suitable to accomplish the task at hand.

L 740-749: Please check the titles of references #42-45.

Thank you for pointing this out. In the original submission, the title for reference #43 was incorrect and now has been fixed. The titles for references 44 and 45 are actually correct – these papers had identical titles despite being separate papers and published in different years! We thank the Reviewer for all the helpful comments and corrections.

REVIEWERS' COMMENTS:

Reviewer #1 (Remarks to the Author):

I think that the manuscript has been revised well to accept for publication.

Reviewer #2 (Remarks to the Author):

Dear Editors, dear Authors,

in my opinion, the manuscript "Organometallic compounds as carriers of extraterrestrial cyanide in primitive meteorites" by Callahan et al. got well-refined and I recommend it for publication in Nature Communications. It's contents is of high interest for a broad scientific community, including the research fields of astrochemistry, astrobiology and cosmochemistry. This work is novel and discusses an important issue, the interaction between organometallic and organic astrochemical compounds. The papers results will potentially influence the research field on the origin(s) of life. The manuscript is well-written, in clear and exciting manner. My decision is based on following aspects:

(i) evidence for nonterrestrial origin
this point got very well-addressed.

(ii) Some statements are speculative
fine. also this issue got addressed and well-refined.

notes within the pdf file

to my full satisfactory, almost all issues have been concerned well. just one point remained: why was NaOH used in sample preparation? i assume, to extract CN most efficiently. this point haven't got answered. here, i am generally a little bit worried if chemical modifications of meteoritic organic matter by pH can be ruled out. nevertheless, in focus of the presented targeted analyte compounds, i am confident that the presented extraction protocol is appropriate for the discussed topic.

Alexander Ruf

Reviewer #3 (Remarks to the Author):

The revision seems to answer well with reviewers' questions and concerns. I recommend for publication.

Reviewer #1 (Remarks to the Author):

I think that the manuscript has been revised well to accept for publication.

Reviewer #2 (Remarks to the Author):

Dear Editors, dear Authors,

in my opinion, the manuscript "Organometallic compounds as carriers of extraterrestrial cyanide in primitive meteorites" by Callahan et al. got well-refined and I recommend it for publication in Nature Communications. Its contents is of high interest for a broad scientific community, including the research fields of astrochemistry, astrobiology and cosmochemistry. This work is novel and discusses an important issue, the interaction between organometallic and organic astrochemical compounds. The papers results will potentially influence the research field on the origin(s) of life. The manuscript is well-written, in clear and exciting manner. My decision is based on following aspects:

(i) evidence for nonterrestrial origin
this point got very well-addressed.

(ii) Some statements are speculative
fine. also this issue got addressed and well-refined.

notes within the pdf file

to my full satisfactory, almost all issues have been concerned well. just one point remained: why was NaOH used in sample preparation? i assume, to extract CN most efficiently. this point haven't got answered. here, i am generally a little bit worried if chemical modifications of meteoritic organic matter by pH can be ruled out. nevertheless, in focus of the presented targeted analyte compounds, i am confident that the presented extraction protocol is appropriate for the discussed topic.

Alexander Ruf

Thank you Dr. Ruf for your comments. With regards to 0.08 M NaOH for sample extraction, we had a few reasons for this: (1) we wanted to extract CN efficiently, (2) we wanted to avoid acidic extraction solvents so that we did not inadvertently release cyanide and therefore lose information about speciation (a short explanation of weak and strong metal-cyanide complexes and pH is provided in Supplementary Note 5), and (3) 0.08 M NaOH was used earlier to trap evolved HCN in solution (during acid digestion and distillation of meteorite samples) and we knew this solvent was compatible with our chemical derivatization reaction in case we wanted to analyze for free/simple cyanides.

Reviewer #3 (Remarks to the Author):

The revision seems to answer well with reviewers' questions and concerns. I recommend for publication.

Thank you to Reviewers #1 and #3 and Dr. Ruf for providing helpful comments, suggestions, and corrections that have helped us improve the quality of our manuscript.